# DSMentor: Enhancing Data Science Agents with Curriculum Learning and Online Knowledge Accumulation

## Abstract

Large language model (LLM) agents have shown promising performance in generating code for solving complex data science problems. Recent studies primarily focus on enhancing in-context learning through improved search, sampling, and planning techniques, while overlooking the importance of the order in which problems are tackled during inference. In this work, we develop a novel inference-time optimization framework, referred to as `DSMentor`, which leverages curriculum learning—a strategy that introduces simpler task first and progressively moves to more complex ones as the learner improves—to enhance LLM agent performance in challenging data science tasks. Our mentor-guided framework organizes data science tasks in order of increasing difficulty and incorporates a growing long-term memory to retain prior experiences, guiding the agent's learning progression and enabling more effective utilization of accumulated knowledge. We evaluate `DSMentor` through extensive experiments on DSEval and QRData benchmarks. Experiments show that `DSMentor` using Claude-3.5-Sonnet improves the pass rate by up to 5.2% on DSEval and QRData compared to baseline agents. Furthermore, `DSMentor` demonstrates stronger causal reasoning ability, improving the pass rate by 8.8% on the causality problems compared to GPT-4 using Program-of-Thoughts prompts. Our work underscores the importance of developing effective strategies for accumulating and utilizing knowledge during inference, mirroring the human learning process and opening new avenues for improving LLM performance through curriculum-based inference optimization.

## 1 Introduction

Data-centric tasks, ranging from statistical analysis to model prediction, are integral to various real-world applications, including healthcare (Miotto et al., 2018), finance (Heaton et al., 2017), and engineering (Chien & Wagstaff, 2017). The ever-evolving demands of data-driven fields call for efficient and robust solutions that can effectively process, interpret, and learn from data.

As large language models (LLMs) demonstrate exceptional capabilities in understanding human-like languages, recent works have leveraged LLMs to solve a wide range of tasks, including text generation (Dathathri et al., 2019), reasoning (Wei et al., 2022), and code generation (Chen et al., 2021b). This has motivated the development of *Data Science agents (DS agents)*—employing LLMs to generate code for data-centric tasks. Thanks to the extensive knowledge built during the training of LLMs, existing DS agents (Cheng et al., 2023; Zhang et al., 2023; 2024a; Hong et al., 2024a) have shown strong capabilities in addressing various data science problems.

However, many data science problems are inherently challenging, often involving vague task descriptions, complex data interpretation, and strict output formats, which results in DS agents failing to solve these problems effectively (Zhang et al., 2024b; Lai et al., 2023; Liu et al., 2024a). Additionally, the knowledge embedded within LLMs is *static*, confined to the data available during their training, posing significant challenges in rapidly evolving fields where up-to-date information is essential for generating accurate and relevant outputs.

To extend the knowledge of LLMs to specific data-centric tasks, recent works (Zhang et al., 2024a; Guo et al., 2024) primarily focus on retrieving external knowledge for solving data science problems,

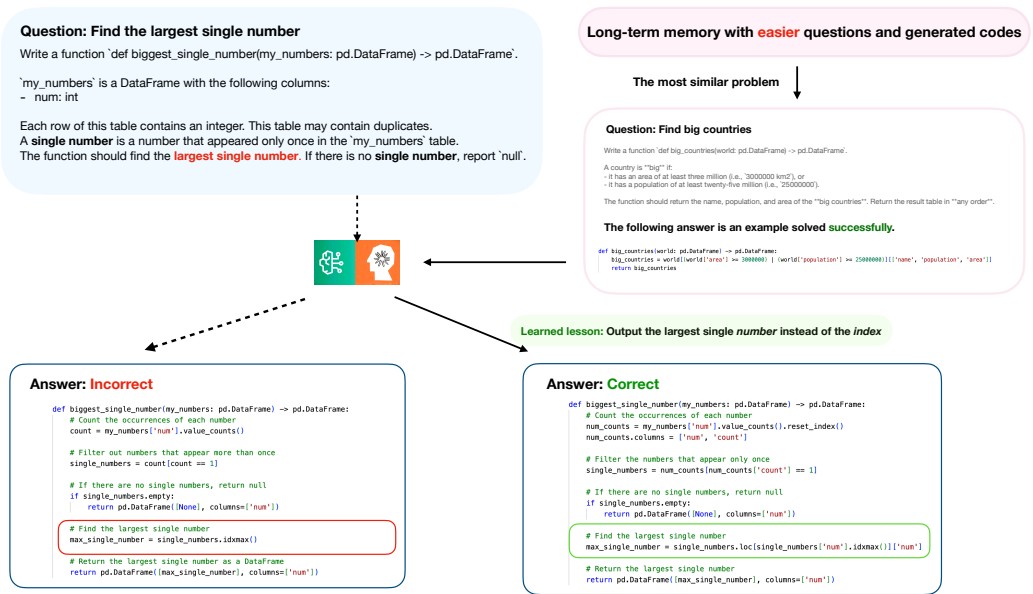

Figure 1: A motivating example illustrating how the agent can learn from previously solved easier problems. Without retrieving information from long-term memory, the agent incorrectly outputs the index of the largest single number, while by leveraging knowledge from previously solved easier questions, the agent can provide the correct answer.

often starting with identifying or collecting relevant knowledge sources. However, these approaches typically do not accumulate knowledge in a dynamic, online fashion, nor do they explore the importance of task order or curriculum in problem-solving. This oversight is particularly critical in data science, where problems are often interrelated, and complex solutions can frequently be constructed by integrating simpler ones (Hong et al., 2024a). For instance, a sophisticated *time series forecasting* task might build upon foundational concepts of *data preprocessing*, *feature engineering*, and basic *regression* techniques. The challenge becomes even more pronounced in online learning or serving scenarios, where the timeliness and quality of the knowledge base directly impact model inference performance. Unfortunately, current methods often result in sub-optimal performance due to inadequate strategies for how to best prepare, structure, and organize the knowledge base.

To address these challenges and leverage the interrelated nature of data science problems, we explore the sequence in which knowledge is populated into and retrieved from a growing memory buffer, helping DS agents better understand and utilize accumulated knowledge. Inspired by the human learning process, where foundational knowledge forms the basis for tackling more complicated problems, we posit that the order in which information is introduced to the model can significantly influence its effectiveness. For instance, understanding basic concepts is often necessary before addressing more advanced challenges, as illustrated in Figure 1.

In this work, we propose a new framework, referred to as DSMentor, which incrementally enhances the capabilities of DS agents through a mentor-guided approach. The *Mentor* agent evaluates the difficulty of all the tasks and provides a sequence order. Specifically, we employ curriculum learning to carefully structure the learning process for DS agents and accumulate knowledge from the solutions to previous tasks. This approach mirrors the hierarchical nature of data science problem-solving, where complex solutions often build upon simpler concepts. By systematically increasing the difficulty of tasks, our framework enables the agent to develop the skills necessary to tackle more complicated challenges within the data science domain. We evaluate our framework using Claude-3.5-Sonnet and Llama-3.1-70b on two popular data science benchmarks: DSEval (Zhang et al., 2024b) and QRData (Liu et al., 2024a).

Our main contributions can be summarized as follows:

- **Curriculum-based performance improvement:** We demonstrate the superior performance of DSMentor, which leverages an easy-to-hard curriculum, across established

benchmarks in data analysis and causal reasoning tasks. Our framework surpasses other data science agents by systematically guiding the learning process, enabling more effective problem-solving, even for complex tasks that require synthesizing advanced knowledge from multiple simpler problems.

- **Enhanced knowledge utilization:** We show that retrieving easier and relevant examples from memory and structuring the knowledge in an increasing-similarity order, significantly improves DS agents' understanding and allows them to more efficiently leverage prior experiences for solving new tasks.

- **Progressive learning for solving advanced causal problems:** We demonstrate how `DSMentor` incrementally introduces tasks with increasing complexity, facilitating the agent's progressive learning of causal relationships. This gradual exposure strengthens its causal reasoning abilities and enables it to effectively address complex data relationships.

## 2 RELATED WORKS

**Curriculum learning.** Inspired by the human learning process, curriculum learning, first introduced by (Bengio et al., 2009), has become a widely-used approach in many applications, including computer vision, natural language processing, and reinforcement learning (Wei et al., 2016; Zhang et al., 2019; Wang et al., 2021; Portelas et al., 2021). This approach involves organizing training examples in a sequence from easier to harder tasks, which facilitates convergence of the training process and improves the quality of the learned models (Hacohen & Weinshall, 2019). Determining task difficulty is a critical factor in enhancing training efficiency (Wang et al., 2021) and can be categorized into pre-defined difficulty (Platanios et al., 2019; Shi et al., 2023; Lu et al., 2024; Liu et al., 2024b) and automatic difficulty measurement (Portelas et al., 2021; Wang et al., 2024; Sun et al., 2024). For instance, the Cross-Episodic Curriculum (CEC) method (Shi et al., 2023) structures learning experiences based on factors such as the pre-defined task difficulty or demonstrator expertise, leveraging cross-episodic attention in Transformers. Recent works (Wang et al., 2024; Sun et al., 2024) employ large language models to automatically generate tasks with a diverse range of difficulties. In this paper, we leverage the LLM-based *Mentor* agent to automatically assess task difficulty and systematically construct a curriculum-based dataset that progresses from simpler to more complex tasks. Instead of focusing on the training stage, this paper incorporates curriculum learning with LLM agents during inference, emphasizing the impact of curriculum on the growth of long-term memory and the examples retrieved from online memory.

**Data science LLM agents.** LLMs have become increasingly valuable in data science, including OpenAI's Codex (Chen et al., 2021a) and Anthropic's Claude (Anthropic, 2023) assisting in data preprocessing, analysis, and code generation through natural language interfaces (Biswas et al., 2023). However, their static nature, where knowledge is fixed at the time of training, limits their ability to handle real-time data. To address this, recently proposed agents such as the Data Interpreter (Hong et al., 2024a), MLCopilot (Zhang et al., 2024a), and AIDE (Schmidt et al., 2024) extend LLM capabilities by incorporating dynamic planning, human-like reasoning, and iterative refinement, significantly enhancing performance in data science problem-solving. Our work builds on these advancements by incorporating curriculum learning and retrieval-based techniques to further enhance LLM agents' adaptability in data science tasks.

**Multi-agent LLM frameworks.** LLMs have seen growing application in multi-agent systems, where multiple agents collaborate to solve complex tasks through communication and cooperation. Frameworks like CAMEL (Li et al., 2023) and MetaGPT (Hong et al., 2024b) demonstrate how multi-agent LLM systems can decompose large projects into modular sub-tasks, allowing specialized agents to handle distinct components of the problem. Moreover, curriculum learning and retrieval-augmented techniques are increasingly integrated into these systems to enhance adaptability and efficiency, as shown in recent work focused on game-play agents (Wang et al., 2024). Additionally, studies on multi-agent debate systems reveal that agent interaction can significantly improve reasoning abilities (Wang et al., 2023; Khan et al., 2024). In this work, we propose a framework where a *Mentor* agent assists a *Student* agent, which enhances the student agent's problem-solving skills in data science tasks.

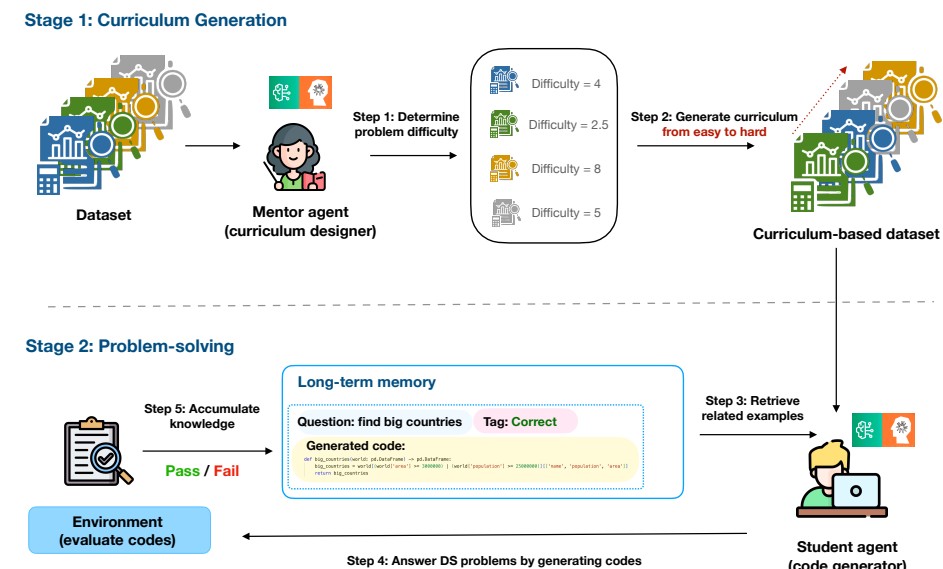

Figure 2: Our framework DSMentor. Here, the *Mentor* agent assesses the difficulty of each problem and generates a curriculum accordingly. Once the curriculum are determined, the *Student* agent—responsible for answering questions through code generation—retrieves relevant examples from an accumulated online long-term memory. After the environment evaluates the generated code, the *Student* agent will append the question, its output and evaluation tag (i.e., incorrect or correct answer), to the long-term memory.

## 3 DSMENTOR: A MENTOR-GUIDED APPROACH

In this section, we present DSMentor, a novel mentor-guided approach that leverages curriculum learning and a growing online memory to enhance the capabilities of DS agents for solving data science tasks. As illustrated in Figure 2, DSMentor operates in two stages: the curriculum-generation stage and the problem-solving stage. In the following subsections, we will discuss the details about these two stages and explain the main components of DSMentor as well as how they work together to facilitate effective learning.

### 3.1 CURRICULUM-GENERATION STAGE

Given any data science task dataset $\mathcal{D}$, we first employ a *Mentor* agent to construct a curriculum-based counterpart $\mathcal{D}_c$. This curriculum establishes the sequence of the tasks that will be undertaken during the problem-solving stage. In this paper, we focus on a difficulty-based curriculum and the preparation of a curriculum-based dataset comprises two key steps.

**Step 1: Determine difficulty.** First, the *Mentor* agent assesses and assigns a difficulty level to each problem. In our setting, we rely only on problem descriptions to gauge the difficulty of each problem, since the ground-truth code or evaluation results are generally not available to the *Mentor* agent during this stage in many real world scenarios. To ensure consistency and scalability across a diverse set of DS tasks, we provide *difficulty scale guidelines* for the *Mentor* agent to analyze and assign difficulty levels. As the examples illustrated in Figure 3, the *Mentor* agent can tell that the task of finding the largest single number is more difficult than identifying big counties based on specific conditions. The *Mentor* agent also provides reasons that the former requires counting occurrences within the data, while the latter only involves filtering a DataFrame satisfying the required conditions.

**Step 2: Generate curriculum.** Once the difficulty level for each task is determined, we structure the curriculum by arranging the tasks in a sequence that progresses from easier to more challenging

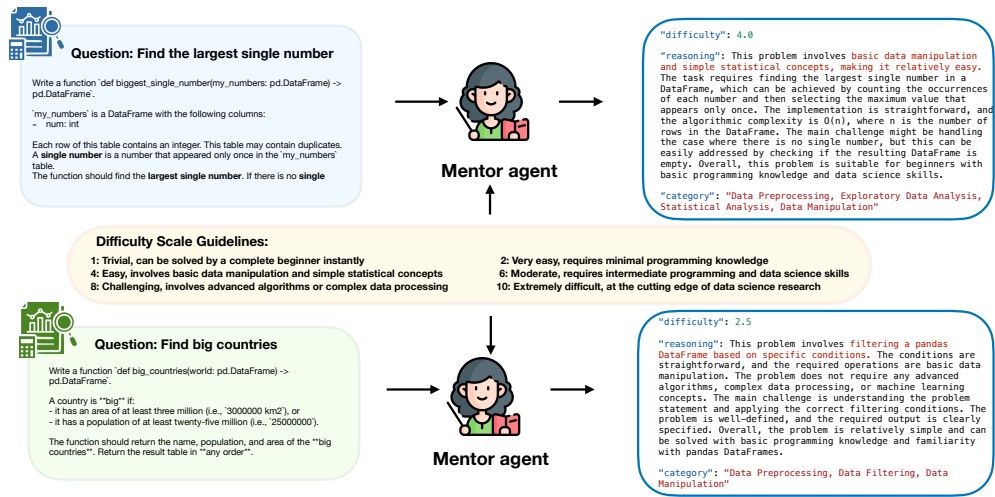

Figure 3: Examples of determining difficulties during the curriculum-generation stage.

problems. Together with the long-term memory described later, this ordering allows the agent to build foundational skills on basic tasks before advancing to more complex ones. For example, as shown in Figure 3, the *Student* agent will first tackle the task of finding big countries, followed by the task of identifying the largest single number, based on their respective difficulty levels.

We also explore the effectiveness of other difficulty metrics and alternative curricula, which will be detailed in 4.4 for comprehensive ablation studies.

### 3.2 PROBLEM-SOLVING STAGE

With the curriculum-based dataset $\mathcal{D}_c$, we utilize a *Student* agent that iteratively solves data science problems, progressing from easier to more complex ones. Inspired by verbal reinforcement learning (Shinn et al., 2024), we introduce a growing *online long-term memory* module that enables the *Student* agent to retain and retrieve knowledge from previously tackled, less complex problems, without necessitating updates to the model weights.

**Online long-term memory:** Assume that there are $N$ problems in the dataset $\mathcal{D}_c$. For each problem $i \in \{1, \ldots, N\}$, we denote its description as $p_i$, where $i$ is the index of $p_i$ in the sequence of problems. The long-term memory can be formally defined as

$$\mathcal{M}_i = \{(p_k, c_k, t_k)\}_{k=1}^{i-1},$$

where $p_k$ and $c_k$ are the problem description and the corresponding generated code of some previous problem $k < i$, and $t_k \in \{\text{Correct}, \text{Incorrect}\}$ is the evaluation tag for problem $k$.

**Step 3: Retrieve from online long-term memory.** While solving the $i$-th problem in the curriculum-based dataset $\mathcal{D}_c$, the *Student* agent can retrieve examples from long-term memory $\mathcal{M}_i$ based on the similarity between the current problem description $p_i$ and previous description $p_k \in \mathcal{M}_i$, according to their cosine similarity: $\text{sim}(p_i, p_k) = \cos(\mathcal{E}(p_i), \mathcal{E}(p_k))$, where $\mathcal{E}$ is the embedding model. We finally retrieve examples with the top-$K$ similarities from the long-term memory, where $K$ is the number of the retrieved examples. The examples include both correctly and incorrectly solved problems along with their corresponding generated code.

**Step 4: Answer DS problem by generating codes.** Using the retrieved examples, the *Student* agent follows the order of increasing similarity. By learning from prior experiences, the *Student* agent attempts to solve the current data science problem by generating code, which is then evaluated by the environment based on its execution output.

**Step 5: Accumulate knowledge to long-term memory.** After obtaining the evaluation results, the *Student* agent appends the question, the generated code, and the evaluation tag (i.e., correct or incorrect) to the long-term memory for future use. As the inference process continues, the long-term memory will keep expanding.

## 4 EXPERIMENTS

In this section, we evaluate the proposed DSMentor on two different data-science benchmarks to address the following research questions: **(Q1)** How does DSMentor compare with other popular data science agents? **(Q2)** Can DSMentor enhance the ability to solve more difficult problems, like causal reasoning? **(Q3)** What is most suitable curriculum design that yields the best performance?

In the sequel, we first attempt to answer **(Q1)** and **(Q2)** in Section 4.3 and conduct a series of ablation studies to explore different curriculum designs in Section 4.4 for **(Q3)**.

### 4.1 EXPERIMENT SETUP

**Benchmarks and datasets.** To answer the aforementioned questions, we conducted experiments on DSEval (Zhang et al., 2024b) and QRData (Liu et al., 2024a), which contains 705 problem-sets with 1236 questions in total. More specifically, DSEval consists of four datasets: LeetCode (with 40 single-turn problem-sets), StackOverflow (with 202 single-turn problem-sets), Exercise (with 21 multiple-turn problem-sets), and Kaggle (with 31 multiple-turn problem-sets), while QRData are composed of 411 statistical and causal reasoning problems. More details are summarized in Appendix A.2 of the supplementary material, where we address certain DSEval evaluation issues that lead to an underestimation of agent capabilities.

**Implementation details.** We equip DSMentor with *anthropic.claude-3-5-sonnet-20240620-v1:0* and *meta.llama3-1-70b-instruct-v1:0* as the base LLMs, referred to as *DSMentor-Claude-3.5-Sonnet* and *DSMentor-Llama-3.1-70b*, respectively. Both the *Mentor* and *Student* agents use the same base model. Moreover, the *Mentor* agent follows the *problem-based difficulty* and the *Student* agent retrieves similar examples from a long-term memory storing previously answered questions and generated codes (*both correct and incorrect*). We employ *cohere.embed-english-v3* as the embedding to retrieve relevant examples from the memory. We set the number of retrieved examples as listed in Appendix A.1 of the supplementary material, which will be discussed with in detail in Section 4.4.3. During inference, we run DSMentor with easy-to-hard or hard-to-easy curriculum for three times and DSMentor with random curriculum for five times to mitigate randomness, reporting average results. The temperature is set to zero for more deterministic behavior.

**Evaluation metrics.** For DSEval, we adopt the *Pass Rate* metric from Zhang et al. (2024b), which is the ratio of correctly answered questions to the total number of questions, without considering any variable violation issues and error propagation for multiple-turn problem-sets. For QRData, we directly evaluate by comparing the execution output of the generated code against the ground-truth results, which can be either numerical or multiple-choice.

### 4.2 BASELINES

We compare DSMentor to the following baseline agents on DSEval (Zhang et al., 2024b) and QRData (Liu et al., 2024a). These agents are selected for their leading performance on the corresponding benchmarks.

**Jupyter-AI** (JupyterLab, 2023) is an open-source tool that enhances Jupyter Notebooks with generative AI capabilities. It demonstrates strong performance on DSEval-LeetCode, outperforming other data science agent frameworks such as Chapyter (chapyter, 2023), Code Interpreter (shroominic., 2023), and CoML (Zhang et al., 2024a).

**CoML (Zhang et al., 2024a)** leverages offline data containing pre-existing tasks and distills the related knowledge to enhance performance during the online stage. It achieves the best performance on DSEval benchmarks, except DSEval-LeetCode compared to other agents (JupyterLab, 2023; chapyter, 2023; shroominic., 2023), which makes it an important baseline for comparison.

| Model (with Llama-3.1-70b) | DSEval-LeetCode | DSEval-SO | DSEval-Exercise | DSEval-Kaggle |
|---|---|---|---|---|
| Jupyter-AI (JupyterLab, 2023) | 0.733 | 0.427 | 0.679 | 0.465 |
| CoML (Zhang et al., 2024a) | 0.683 | 0.794 | **0.783** | **0.577** |
| Llama-3.1-70b (PoT) | 0.725 | 0.739 | 0.747 | 0.539 |
| **DSMentor-Llama-3.1-70b** | **0.792** | **0.837** | 0.752 | **0.577** |
| **Model (with Claude-3.5-Sonnet)** | **DSEval-LeetCode** | **DSEval-SO** | **DSEval-Exercise** | **DSEval-Kaggle** |
| Jupyter-AI (JupyterLab, 2023) | 0.850 | 0.477 | 0.693 | 0.595 |
| CoML (Zhang et al., 2024a) | 0.875 | 0.809 | 0.643 | 0.535 |
| Claude-3.5-Sonnet (PoT) | 0.800 | 0.804 | 0.745 | 0.641 |
| **DSMentor-Claude-3.5-Sonnet** | **0.892** | **0.837** | **0.781** | **0.678** |

Table 1: Performance comparison of our `DSMentor` and existing data science models across DSEval-LeetCode, DSEval-SO, DSEval-Kaggle, and DSEval-Exercise.

| Model | Pass Rate | Multiple Choice/Numerical | Statistical/Causal |
|---|---|---|---|
| GPT-4 (PoT) | 0.491 | 0.460/**0.540** | **0.725**/0.368 |
| Llama-3.1-70b (PoT) | 0.442 | 0.480/0.384 | 0.615/0.351 |
| Claude-3.5-Sonnet (PoT) | 0.471 | 0.495/0.436 | 0.646/0.379 |
| **DSMentor-Llama-3.1-70b** | 0.508 | 0.566/0.419 | 0.676/**0.419** |
| **DSMentor-Claude-3.5-Sonnet** | **0.543** | **0.602**/0.452 | 0.707/**0.456** |

Table 2: Performance comparison of our `DSMentor` and existing data science models on QRData, which includes the overall pass rate, along with breakdowns for multiple choice/numerical and statistical/causal questions.

**Vanilla agents with Program-of-Thoughts (PoT) (Chen et al., 2023)** solve tasks by generating Python code, with the executed output serving as the solution. For each benchmark, Llama-3.1-70b and Claude-3.5-Sonnet act as vanilla agents, using the same system instructions as `DSMentor` to generate code (see Appendix A.3.1 in the supplementary material). Additionally for QRData, we include GPT-4 (Cheng et al., 2023) with PoT as one of the baseline agents, as it performs best among PoT-style agents reported in Liu et al. (2024a). To ensure a fair and consistent comparison, we focus on single-turn code generation alternatives rather than ReAct-style (Yao et al., 2022) or other agents that involve multi-turn reasoning and error feedback.

## 4.3 RESULTS AND DISCUSSION

**Baselines vs DSMentor.** As shown in Table 1 and 2, DSMentor-Claude-3.5-Sonnet consistently outperforms baseline agents across multiple datasets. Compared to prior art Jupyter-AI and CoML, it achieves significant improvements, including an **8.8% increase** on DSEval-Exercise and and **8.3% boost** on DSEval-Kaggle. DSMentor-Llama-3.1-70b also shows notable gains, with an **5.9% improvement** on DSEval-LeetCode and a **4.3% improvement** on DSEval-SO. Despite performing slightly worse on DSEval-Exercise compared to CoML with Llama-3.1-70b, `DSMentor` demonstrates strong performance on QRData as shwon in Table 2, which contains larger and more difficulty problem sets.

**Online long-term memory.** Compared to the vanilla Llama-3.1-70b and Claude-3.5-Sonnet using PoT prompts, `DSMentor`, equipped with the same LLM, significantly improves the pass rate on both DSEval and QRData. Such improvements are driven by the presence of online long-term memory and curriculum learning. Furthermore, they are especially noticeable on more difficult datasets, such as DSEval-Kaggle and QRData, where leveraging previous examples plays a crucial role in enhancing performance.

**Causal reasoning tasks.** Causal reasoning is considered more challenging than other statistical problems in QRData (Liu et al., 2024a). From Table 2, DSMentor-Claude-3.5-Sonnet achieves 45.6%, which improves GPT-4 (PoT) as the best baseline model by 8.8% on the causal problems. Such an improvement indicates that accumulating knowledge from easier questions is particularly effective for complex reasoning, especially in cases where understanding causal relationships is crucial.

| Difficulty Measurement | DSEval-LeetCode | DSEval-SO |
|---|---|---|
| Manual | 0.742 | – |
| Reference-code-based | 0.750 | 0.812 |
| Pass-rate (Llama-3.1-8b) | **0.800** | 0.825 |
| Pass-rate (Claude-3-Haiku) | 0.775 | 0.812 |
| Problem-based only | 0.792 | **0.837** |
| *baseline w/o curriculum:* Llama-3.1-70b (PoT) | 0.725 | 0.739 |

Table 3: Comparison among different difficulty designs for DSMentor-Llama-3.1-70b, where the numbers of retrieved examples are 5 and 15 for DSEval-LeetCode and DSEval-SO respectively. We also report baseline numbers without curriculum in the last row.

| Difficulty Measurement | DSEval-LeetCode | DSEval-SO |
|---|---|---|
| Manual | 0.892 | – |
| Reference-code-based | 0.825 | 0.832 |
| Pass-rate (Llama-3.1-8b) | 0.883 | 0.827 |
| Pass-rate (Claude-3-Haiku) | 0.867 | 0.815 |
| Problem-based only | **0.892** | **0.837** |
| *baseline w/o curriculum:* Claude-Sonnet-3.5 (PoT) | 0.800 | 0.804 |

Table 4: Comparison among different difficulty designs for DSMentor-Claude-3.5-Sonnet, where the number of retrieved examples is 5 for both DSEval-LeetCode and DSEval-SO.

In summary, DSMentor, incorporating with curriculum learning and long-term memory, shows a noticeable advantage over baseline models, particularly on complex datasets like DSEval and QRData. Our results underscore the importance of scalable curriculum strategies and long-term memory in improving AI-assisted data science performance.

## 4.4 ABLATIONS

Next, we conduct a series of ablation studies to examine the impacts of each components of DSMentor, regarding curriculum and long-term memory designs.

### 4.4.1 DIFFICULTY DEFINITION

Figuring out an appropriate difficulty measurement plays a critical role in our framework. In addition to the problem-based difficulty used in Section 4, where the LLM-based Mentor agent assesses difficulty based solely on each question, we evaluate three additional difficulty metrics and analyze their performance on DSEval-LeetCode and DSEval-SO. Below, we summarize the key concepts of these difficulty metrics, where more details are postponed to Appendix B.

**Manual difficulty (for DSEval-LeetCode only):** referring to the difficulty level (easy/medium/hard) as defined on LeetCode. Problems are first sorted by their original difficulty level, and in case of a tie, human pass rate is used to further rank them (i.e., a higher pass rate indicates an easier problem).

**Reference-code-based difficulty:** following the difficulty in DSEval (Zhang et al., 2024b), based on the number of functions, variables, conditions, and loops in the reference code.

**Pass-rate difficulty:** employing the pass rate of weaker models, such as *Llama-3.1-8b* and *Claude-3-Haiku*, where a higher pass rate indicates an easier question.

To explore the impact of different difficulty definitions, we follow the same experiment setup described in Section 4.1 except the curriculum generation process that utilizes different difficulty metrics. We run each experiment for three times and present the average result in Table 3 and 4 to mitigate the randomness. As shown in Table 3 and 4, we first observe that incorporating with the curriculum designs significantly improves the overall pass rate for both DSMentor-Llama-3.1-70b and DSMentor-Claude-3.5-Sonnet on DSEval-LeetCode and SO, comparing to the baseline vanilla agent without memory and curriculum. Moreover, the problem-based difficulty generally

| Curriculum Design | DSEval-LeetCode | DSEval-SO | DSEval-Exercise | DSEval-Kaggle |
|---|---|---|---|---|
| Easy-to-Hard (Inc. Similarity) | **0.792** | **0.837** | **0.752** | 0.577 |
| Easy-to-Hard (Inc. Difficulty) | 0.775 | **0.837** | 0.729 | 0.566 |
| Hard-to-Easy (Inc. Similarity) | **0.792** | 0.807 | 0.740 | **0.583** |
| Random (Inc. Similarity) | 0.770 | 0.803 | 0.750 | 0.571 |

Table 5: Performance comparison of DSMentor-Llama-3.1-70b with different curriculum designs across DSEval. Here, Inc. Similarity and Inc. Difficulty represent that the retrieved examples are in the order of increasing similarity or increasing difficulty, respectively.

| Curriculum Design | DSEval-LeetCode | DSEval-SO | DSEval-Exercise | DSEval-Kaggle |
|---|---|---|---|---|
| Easy-to-Hard (Inc. Similarity) | **0.892** | **0.837** | **0.781** | 0.678 |
| Easy-to-Hard (Inc. Difficulty) | 0.850 | 0.835 | 0.774 | **0.684** |
| Hard-to-Easy (Inc. Similarity) | 0.833 | 0.797 | 0.742 | 0.680 |
| Random (Inc. Similarity) | 0.850 | 0.800 | 0.775 | 0.678 |

Table 6: Performance comparison of DSMentor-Claude-3.5-Sonnet with different curriculum designs across DSEval.

outperforms the other difficulty metrics, although DSMentor-Llama-3.1-70b using the problem-based difficulty performs slightly worse than when using the pass-rate generated by Llama-3.1-8B on DSEval-LeetCode. We further examine the correlation between problem-based difficulty and pass-rate difficulty that exclusively reflects each model's problem-solving abilities for each tasks. Due to page limits, we postpone the detailed discussion and results to Appendix D.

### 4.4.2 Order of tasks and examples

In addition to difficulty metrics, we further explore the impact of different task orders and retrieved examples, both of which are key components of curriculum design.

First, we compare the easy-to-hard curriculum with the random and hard-to-easy curriculum, where the retrieved examples are ordered by increasing similarity (denoted as *Inc. Similarity*). As shown in Table 5 and 6, both DSMentor-Llama-3.1-70b and DSMentor-Claude-3.5-Sonnet perform better with the easy-to-hard or hard-to-easy curricula compared to the random curriculum, demonstrating the efficacy of curriculum learning. Additionally, the easy-to-hard curriculum often shows better performance than the hard-to-easy curriculum, except for DSEval-Kaggle, where the hard-to-easy curriculum performs slightly better by 0.4%.

Given that curriculum design influences performance through retrieved examples, we also investigate different ranking approaches for these examples. In addition to increasing similarity, we use increasing difficulty (denoted as *Inc. Difficulty*) for easy-to-hard curriculum. As seen in Table 5 and 6, our experimental results show that the easy-to-hard curriculum with examples ordered by increasing similarity often outperforms alternative approaches, except on DSEval-Kaggle.

### 4.4.3 Long-term memory

As the *Student* agent utilizes examples retrieved from long-term memory, we then examine the impact of long-term memory, specifically focusing on the number of retrieved examples and the inclusion of incorrectly answered question with failed attempts.

**The impact of the number of retrieved examples.** We first vary the number of retrieved examples and examine its impact for both DSMentor-Llama-3.1-70b and DSMentor-Claude-3.5-Sonnet over all the five datasets. Except the number of examples, all setups follow Section 4.1. From Figure 4, DSMentor with retrieved examples (i.e. # examples > 0) consistently improves performance across all datasets. However, we observe that DSMentor-Llama-3.1-70b often experiences a performance drop as the number of retrieved examples increases, particularly on multi-turn datasets such as DSEval-Exercise and DSEval-Kaggle. In contrast, DSMentor-Claude-3.5-Sonnet generally

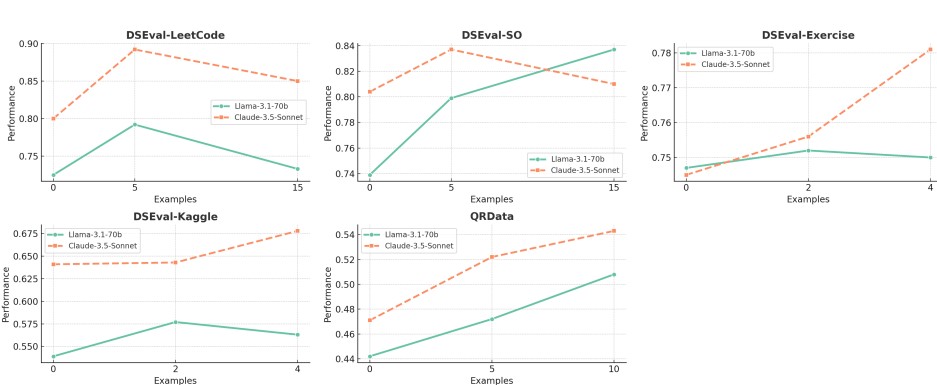

Figure 4: Performance of DSMentor models across different datasets on DSEval and QRData, with varying number of retrieved examples. The subfigures show the results for each dataset, demonstrating performance trends for Llama-3.1-70b and Claude-3.5-Sonnet models.

| Dataset | Incorrect Examples | Pass Rate |
|---|---|---|
| DSEval-LeetCode | ✗ | 0.733 |
| | ✓ | **0.792** |
| DSEval-SO | ✗ | 0.830 |
| | ✓ | **0.837** |
| DSEval-Exercise | ✗ | 0.725 |
| | ✓ | **0.752** |
| DSEval-Kaggle | ✗ | 0.562 |
| | ✓ | **0.577** |
| QRData | ✗ | 0.475 |
| | ✓ | **0.508** |

Table 7: DSMentor-Llama-3.1-70b with and without incorrectly answered examples.

benefits from additional retrieved examples, often demonstrating improved performance with more retrieved examples.

**The impact of adding incorrect examples.** Next, we further explore whether the *Student* agent could benefit from incorrect examples (i.e., previously answered questions that are incorrect). We evaluate DSMentor-Llama-3.1-70b on DSEval and QRData, both with and without incorrect examples retrieved from the long-term memory. Apart from incorporating incorrect examples, all experimental setups follows Section 4.1. Table 7 shows that enabling access to incorrect examples can enhance the pass rate, especially for datasets with fewer problem-sets (e.g., DSEval-LeetCode and DSEval-Exercise), though the datasets with a larger number of problem-sets (e.g., DSEval-SO, DSEval-Kaggle and QRData), the improvement from adding incorrect examples is marginal. This observation aligns with the principle that larger datasets generally possess a more extensive memory pool for retrieval, thereby increasing the likelihood of selecting sufficient correct examples without necessitating additional learning signals from incorrect instances.

## 5 CONCLUSION

In this work, we develop `DSMentor`, a framework that enhances data science agents through a mentor-guided curriculum learning approach, effectively improving their ability to tackle complex data science tasks. Extensive experiments demonstrate that organizing tasks from easy to hard significantly boosts the agent's problem-solving and causal reasoning capabilities by building a strong knowledge foundation during inference. Future directions include exploring adaptive curriculum that adjust task difficulty based on agent performance, incorporating advanced memory mechanisms for better knowledge retention, and extending `DSMentor` to support multi-agent collaboration in complex, interdisciplinary domains.

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

## A ADDITIONAL EXPERIMENTAL DETAILS

### A.1 THE NUMBER OF RETRIEVED EXAMPLES

In Table 8, we summarize the number of retrieved examples used in our main experiments.

| Dataset | DSMentor-Claude-Sonnet-3.5 | DSMentor-Llama-3.1-70b |
|---|---|---|
| DSEval-LeetCode | 5 | 5 |
| DSEval-SO | 5 | 15 |
| DSEval-Exercise | 2 | 4 |
| DSEval-Kaggle | 2 | 4 |
| QRData | 10 | 10 |
| QRData-Causal | 10 | 10 |
| QRData-Stats | 10 | 10 |

Table 8: The number of retrieved examples used in Section 4 for each dataset.

### A.2 DATASET AND BENCHMARK CORRECTION

In DSEval (Zhang et al., 2024b), the `TableTestValidator` is used to evaluate questions about function implementation using multiple test-cases, which may also include the checker of the function input. This validator is mostly used for DSEval-LeetCode. If there is no requirement for input checker, DSEval by default will initialize as

**Initialization Issue of `input_checker`**

```python
# Initialization for TableTestValidator
def __init__(
    self,
    ...,
    input_checker: str | CompareFunction | dict | bool | None = None):

    if input_checker is False:  # <- This condition is problematic
        self.input_checker = None
    else:
        if input_checker in (True, None):
            input_checker = None  # Default handling
        self.input_checker = (
            _compare_fn_from_config(input_checker),
            _compare_fn_from_config(input_checker, loose=True),
        )
    # Other initialization
    ...
```

The above code always follows the `branch` and sets a `input_checker`. To address this issue, we initialize as `input_checker = False`.

In addition, we observed that most agents failed on the `trips-and-users` problem due to an unclear format requirement in the problem description. Specifically, the string-based problem description does not explicitly specify the expected format of the date output.

**Format Issue with `output_checker`**

```python
# Original version
def compare_fn(expected, output):
    if not np.issubdtype(output['Cancellation Rate'], np.number):
```

```
        output['Cancellation Rate'] =
    ↪   pd.to_numeric(output['Cancellation Rate'])
    return {
        'match': expected.sort_values(by=['Day', 'Cancellation
    ↪   Rate']).reset_index(drop=True).equals(output.sort_values(
    ↪   by = ['Day', 'Cancellation
    ↪   Rate']).reset_index(drop=True)),
        'reason': ''
    }
# Modified version
def compare_fn(expected, output):
    if not np.issubdtype(output['Cancellation Rate'], np.number):
        output['Cancellation Rate'] =
    ↪   pd.to_numeric(output['Cancellation
    ↪   Rate']).astype(float)
    if not np.issubdtype(output['Day'], np.datetime64):
        output['Day'] = output['Day'].astype(np.datetime64)
    return {
        'match': expected.sort_values(by=['Day', 'Cancellation
    ↪   Rate']).reset_index(drop=True).equals(output.sort_values(
    ↪   by = ['Day', 'Cancellation
    ↪   Rate']).reset_index(drop=True)),
          'reason': ''
    }
```

In the modified version, we ensure that the output is consistently formatted, allowing it to be accurately compared with the ground truth, which mitigates the unclear format description for datetime output.

### A.3 PROMPT DESIGNS

In this section, we provide detailed explanations of our prompt designs for the purpose of reproduction. For system message, it consists of system instructions and guidelines (described in Appendix A.3.1), as well as long-term memory prompts, if applicable (described in Appendix A.3.2).

### A.3.1 SYSTEM INSTRUCTIONS AND GUIDELINES

We first presents the system message of DSMentor-Claude-3.5-Sonnet as a follow.

---

**System Instructions and Guidelines**

You are a data scientist assistant skilled in writing Python code for data analysis, visualization, and machine learning tasks. You can leverage Python libraries like pandas, scikit-learn, matplotlib, seaborn, etc. to fulfill the user's requests.

The user will present a question and may optionally provide variables (e.g., pandas DataFrame). Your task is to write a cell in an iPython notebook that addresses the user's request. The cell's output should be exactly what the user has asked for.

## Instruction and guidelines

- Wrap the generated code with ``` before and with ``` after it.

- Import necessary libraries at the beginning.

- Ensure that the executing code only calls functions and uses the variables that are defined properly or imported within the available scope.

- Ensure that executing the code generates the desired output. Do not use `# print` or `return` statements to generate the result outside the function if any. For non-function-completion questions, ```var_a = xxx; var_a``` will generate the desired output of var_a and the output of a cell is the last statement of the code.

---

- Important: Do not overwrite or modify the variables provided by the user, unless explicitly asked to do so. For example, if the user provides a DataFrame df, do not reassign df unless the user requests modifying it in-place.

- Write runnable code directly, without providing examples.

- [For DSEVal-LeetCode only] If the question explicitly asks to write a function, you need to use ```` ```return xxx``` ```` only within the function scope to ensure the output. Outside the function, do not include any example or ```` ```print``` ```` statements to generate the output.

- [For QRData only] For multiple choice questions, store the generated results in ```` ```selected_choice``` ```` and the last cell will be ```` ```selected_choice``` ```` to ensure an output.

- [For multiple-turn datasets only] Use only the provided variables from the current environment (listed after ```` ```Variables:``` ````) to solve the given question.

- [For multiple-turn datasets only] Do not any unnecessary operations and content, such as sorting data, including examples or usage instructions, unless explicitly required by the question.

- [For multiple-turn datasets only] Outside the function, do not include any example or ```` ```print``` ```` statements to generate the output.

As for DSMentor-Llama-3.1-70b, the system message is slightly different, listed as follows:

- The 1st guideline: Wrap the generated code with ```` ```python ```` before and ```` ``` ```` after the code.

- The 3rd guideline: Ensure that the executing code only calls functions that are defined or imported within the available scope or use the defined variables or the existing keys of imported data properly.

- The 4th guideline: The output of a cell is the last statement of the code. Ensure that executing the code generates the desired output. If it is not within a function, do not use `# print` or `return` statements to output the result. For example, ```` ```var_a = xxx; var_a ```` will generate the desired output of `var_a`.

- For DSEval-LeetCode only, add one instruction:

  - Your solution should follow the format in provided examples and use the schemas provided in the variable/table description.

### A.3.2 LONG-TERM MEMORY PROMPTS

After retrieving examples, we add them to system message in the following ways.

**Long-term Memory Prompts with $K$ Examples**

```
<long-term memory>
```

These are the top-$K$ most similar problems, ordered from least similar to most similar/easy to hard problems. Please utilize the experience you have gained here to help you with the current problem. More specifically, summarize what you can learn from the correct answers (within `<correct_answer> ... </correct_answer>`) and how to avoid the mistakes seen in the incorrect examples (within `<wrong_answer> ... </wrong_answer>`) before generating your code.

```
<examples>
<example>
<type>
```
{correct/wrong answer}
```
</type>
```

```
<question>
{Here is the problem description.}
<variable_context>
{Here is the variable description if any.}
</variable_context>
</question>
The following answer is an example that you have successfully solved/answered incorrectly.
<correct/wrong_answer>
```python
{Here is the generated codes.}
```
</correct/wrong_answer>
</example>
...
</examples>
</long-term memory>
```

## B  DIFFICULTY DESIGNS

In this section, we provide details on various approaches to generating difficulty.

### B.1  PROBLEM-BASED DIFFICULTY

To begin, we present the system message and problem-prompt template that the Mentor agent follows in main experiments.

---

**System Message (Guidelines)**

You're an experienced data scientist with expertise in assessing the difficulty. Your goal is to provide accurate and diverse assessments across different problems. Use the full range of the difficulty scale to differentiate between problems effectively.

Difficulty Scale Guidelines:
1: Trivial, can be solved by a complete beginner instantly
2: Very easy, requires minimal programming knowledge
4: Easy, involves basic data manipulation and simple statistical concepts
6: Moderate, requires intermediate programming and data science skills
8: Challenging, involves advanced algorithms or complex data processing
10: Extremely difficult, at the cutting edge of data science research

Feel free to use any float value between 1 and 10 for precise assessments. Don't hesitate to use the full range when appropriate.

---

**User Message (Problem Prompts)**

Analyze the following coding problem:

Problem:
{Here is the problem description.}

Please provide:
1. A detailed reasoning about the difficulty of this problem (consider concepts involved, algorithmic complexity, implementation challenges, etc.)
2. A difficulty score from 1 to 10, where 1 is very easy and 10 is extremely difficult.
3. Category tags for this problem:

---

> - Generate at least 3 and up to 5 relevant tags for this problem.
>
> - Include both general and specific tags as appropriate.
>
> - You may use tags from the following list, but also feel free to create your own: "Data Preprocessing", "Exploratory Data Analysis", "Machine Learning", "'Deep Learning", "Natural Language Processing", "Computer Vision", "Time Series Analysis", "Statistical Analysis"
>
> - Rank the tags by their correlation to the problem, with the most relevant tag first.
>
> Please output your response following this format:
> <reason>
> [Your detailed reasoning here]
> </reason>
> <difficulty score>
> [Your score as any float between 1 and 10]
> </difficulty score>
> <category>
> [Tag1], [Tag2], [Tag3], [Tag4 (if applicable)], [Tag5 (if applicable)]
> </category>
>
> Ensure that the tags are listed in order of relevance, separated by commas.

We further extract the `difficulty score` as the reference for curriculum generation. In our experiments, we generate difficulty scores three times and take the average as the difficulty level.

## B.2  MANUAL DIFFICULTY

For DSEval-LeetCode, we collect the official difficulty levels from *LeetCode*[1], which are *Easy*, *Medium*, and *Hard*. Within each difficulty level, we further differentiate tasks based on their pass rates as of July 22, 2024.

## B.3  REFERENCE-CODE-BASED DIFFICULTY

As provided by DSEval (Zhang et al., 2024b), reference-code-based difficulty refers to code complexity quantified by factors such as the number of functions, variables, conditions, and loops. We have extracted the relevant code for readers' reference as follows.

**Reference-code-based Difficulty (Zhang et al., 2024b)**

```python
def get_code_complexity(code: str) -> float:
    import ast

    module = ast.parse(code)

    complexity = 0.0
    for node in ast.walk(module):
        # Conditions
        if isinstance(node, (ast.For, ast.While, ast.If,
        ↪  ast.With)):
            complexity += 3
        # Other statements
        elif isinstance(node, ast.stmt):
            complexity += 1
        # Constant, variable
        elif isinstance(node, (ast.Constant, ast.Name)):
            complexity += 0
        # Call other methods
```

---

[1]LeetCode: https://leetcode.com

```
        elif isinstance(node, ast.Call):
            complexity += 3
        # Calling an attribute or a subscript
        elif isinstance(node, (ast.Attribute, ast.Subscript)):
            complexity += 4
        # Other expressions
        elif isinstance(node, ast.expr):
            complexity += 1
        # Function arguments
        elif isinstance(node, ast.arg):
            complexity += 1

    return complexity
```

### B.4 PASS-RATE DIFFICULTY

In this approach, we select vanilla Llama-3.1-8B and Claude-3-Haiku as the weaker agents, using the same system instructions as outlined in Appendix A.3.1 , but without utilizing long-term memory. For each problem, the agent is given up to five attempts to provide a correct answer. We set the temperature to 0.5, which encourages the generation of a diverse range of responses. To assess the difficulty of the tasks, we employ a discounted pass-rate approach: for each problem $k$, the pass-rate difficulty is defined by

$$\texttt{pass-rate difficulty}(k) = 1 - \gamma^{t_k - 1},$$

where the discount factor $\gamma$ is set to 0.9, and $t_k$ represents the number of trials until a correct answer is given or the maximum number of attempts is reached for problem $k$. This formulation implies that a higher pass-rate difficulty score signifies a more challenging problem, as it takes more attempts for the agent to provide a correct answer. Conversely, if the agent solves the problem in fewer attempts, resulting in a lower pass-rate difficulty, it indicates that the question is easier.

## C    PERFORMANCE ON QRDATA-CAUSAL

To demonstrate the causal reasoning abilities of `DSMentor`, we evaluate the performance of different curriculum designs in solving causal reasoning problems from QRData, referred to as QRData-Causal. The results are summarized in the following figure.

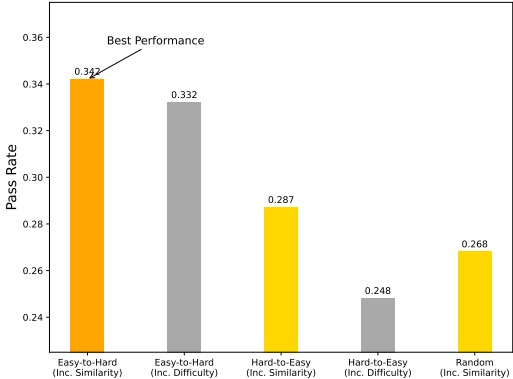

Figure 5: Performance of DSMentor-Llama-3.1-70b on QRData-Causal.

The results suggest that an easy-to-hard curriculum, with increasing similarity as the ranking approach, may be better suited for causal reasoning tasks.

# D    CORRELATION BETWEEN PROBLEM-BASED AND PASS-RATE DIFFICULTY

To better understand the relationship between the problem-based and pass-rate difficulty metrics, we analyze their correlation using both linear (Pearson) and non-linear (Spearman) correlation coefficients, on DSEval-LeetCode. Figure 6 presents the correlation analysis between problem-based and pass-rate difficulty scores across different problems.

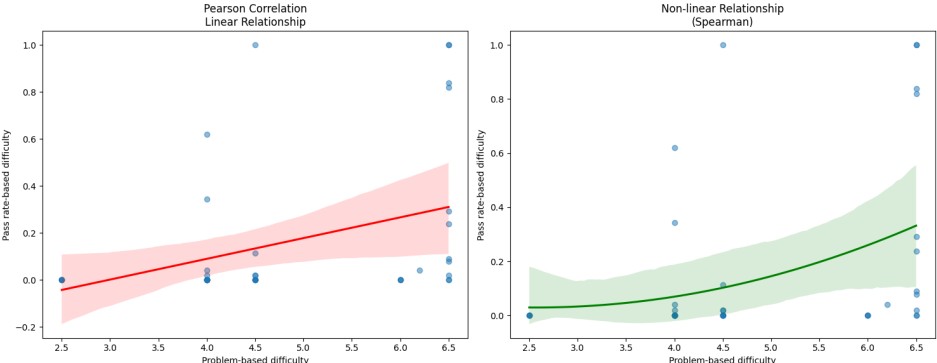

(a) Linear and non-linear correlation analysis between problem-based difficulty and pass rate for Llama-3.1-70b (Pearson correlation: 0.364, $p < 0.05$; Spearman correlation: 0.480, $p < 0.01$).

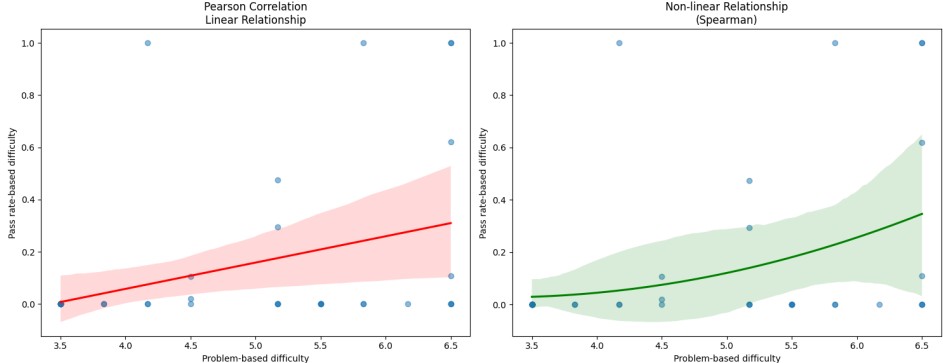

(b) Linear and non-linear correlation analysis between problem-based difficulty and pass rate for Claude-3.5-Sonnet (Pearson correlation: 0.365, $p < 0.05$; Spearman correlation: 0.395, $p < 0.05$).

Figure 6: Correlation between pass-rate and problem-based difficulty on DSEval-LeetCode. For each model, the left plots show Pearson correlation (linear relationship), while the right plots show Spearman correlation (non-linear relationship).

The results show a moderate positive correlation between the two difficulty metrics. For example, for Llama-3.1-70b, the Pearson correlation coefficient of 0.364 ($p < 0.05$) indicates a weak to moderate linear relationship. Meanwhile, the Spearman correlation coefficient of 0.480 ($p < 0.01$) points to a slightly stronger monotonic relationship, implying that non-linear analysis may better capture the relationship between these difficulty metrics.

Notably, there is significant scatter in the data points, particularly for problems with higher difficulty scores (4.0-6.5), where pass rates vary widely from 0% to 100%. This dispersion in the data, despite the statistically significant correlations, suggests that while these difficulty metrics are related, they likely capture different aspects of problem complexity. The moderate correlation also helps explain why different difficulty metrics showed varying performance in our earlier experiments (Tables 3 and 4), highlighting the value of considering multiple approaches to difficulty assessment in curriculum design in the future.

