# OpenReview forum: "DSMentor: Enhancing Data Science Agents with Curriculum Learning and Online Knowledge Accumulation"
_ICLR.cc/2025/Conference — Submitted to ICLR 2025_

### Official Review · Reviewer_XQT5 · 2024-10-26

**Soundness:** 1
**Presentation:** 3
**Contribution:** 1
**Rating:** 3
**Confidence:** 4

**Summary:**

This paper investigates how the order of the problems affects the performance of data science LLM agents. Specifically, this paper proposes DSMentor, which first prompts LLMs to identify the difficulty of the problems in the dataset, and then solves them in the sequence of increasing difficulty on top of RAG. Experiments on DSEval and QRData demonstrate the effectiveness of the proposed framework.

**Strengths:**

- The presentation quality is good.
- The experiments are diverse.
- Open-sourced LLMs are considered for empirical evaluation.

**Weaknesses:**

1. The investigated setting seems unrealistic for me. There is rarely a practical scenario where a set of data science problems with diverse difficulties are given at the same time. For example, in industrial scenarios, the problems with different degrees of difficulties typically come one by one. Moreover, the executability (where LLM agents can obtain the realistic execution feedback) is the most informative feature in data science tasks. However, the setting investigated in this paper fully ignores this, further making the setting more unrealistic.

2. The contribution of this paper is limited. From the perspectives of techniques, I think this paper does not present sufficient novel insights.

- The proposed method of estimating the difficulty is trivial and resource-consuming: 1) How does the difficulty estimation of LLMs align with the ground-truth results? (Results of Pass-rate in Table 3 could have demonstrated this; however, it utilizes llama-8b instead of llama-70b, which fails to fully present this. ) 2) If the estimation is not precise enough, how does it affect the performance for DSMentor? 3) How many tokens (or time/computation resources) are consumed to estimate the difficulties for the whole dataset? Is this worthy enough for performance improvement?

- The order of retrieved examples in RAG has been already investigated in prior works (e.g., [1]); thus, it is not new for the community.

[1] Wang X, Wang Z, Gao X, et al. Searching for best practices in retrieval-augmented generation[J]. arXiv preprint arXiv:2407.01219, 2024.

- The most exciting part of data science task is machine learning, which involves lots of details such as feature engineering, model selection, hyper-parameter tuning. However, this paper seems not to report any results on this.

3. In Line 302-304, why is DSMentor run with three times? If temperature is set to 0 (which means greedy encoding), I cannot identify any randomness involved in this process.

4. In Table 5 and Table 6, it seems that Random (Inc. Similarity) achieves competitive results compared with DSMentor, while avoiding extra inference computation for estimating difficulties.

**Questions:**

How does DSMentor with relatively weak LLMs perform (such as llama-3.1-8B-instruct)?

---

> ### Author Response · Authors · 2024-12-01
> **Response to Reviewer XQT5 [Part 1]**
>
> Thanks for the reviewer’s valuable comments and the acknowledgement on the presentation of this paper and the diverse experimental results. We carefully reviewed all comments and addressed the concerns on a point-by-point basis, as follows.
>
> > [W1] *The investigated setting seems unrealistic for me. There is rarely a practical scenario where a set of data science problems with diverse difficulties are given at the same time. For example, in industrial scenarios, the problems with different degrees of difficulties typically come one by one. Moreover, the executability (where LLM agents can obtain the realistic execution feedback) is the most informative feature in data science tasks. However, the setting investigated in this paper fully ignores this, further making the setting more unrealistic.*
>
> Thank you for raising the concern on applicability. Our current framework primarily focuses on the mentor-guided structure and the sequence of tasks that will be tackled during inference. However, to manage the scenarios with incoming tasks during inference, the Mentor agent can still estimate the difficulty of the new ones and the Student agent can solve them in the order of increasing difficulty,  by retrieving available examples from long-term memory that are simpler than the current task. Regarding executability feedback, our work currently allows the Student agent to receive feedback on the correctness of its responses. However, we acknowledge that integrating detailed execution feedback for code refinement could enhance our model further. We believe this is a valuable direction for future enhancements.
>
> > [W2] *The contribution of this paper is limited. From the perspectives of techniques, I think this paper does not present sufficient novel insights. The proposed method of estimating the difficulty is trivial and resource-consuming: 1) How does the difficulty estimation of LLMs align with the ground-truth results? (Results of Pass-rate in Table 3 could have demonstrated this; however, it utilizes llama-8b instead of llama-70b, which fails to fully present this. ) 2) If the estimation is not precise enough, how does it affect the performance for DSMentor? 3) How many tokens (or time/computation resources) are consumed to estimate the difficulties for the whole dataset? Is this worthy enough for performance improvement?*
>
> Thanks for your concern. We would like to clarify that
>
> 1) We investigated the performance of problem-based difficulty, which utilizes the difficulty scale guidelines, compared with other difficulty metrics in Section 4.4.1. From Table 3 and 4, we observe that using the problem-based difficulty **outperforms that with the manual difficulty** (based on the official difficulty and pass rate provided by LeetCode) by 5% for DSMentor-Llama-3.1-70b and achieves the same pass rate for DSMentor-Claude-3.5-Sonnet, which substantiates the effectiveness of difficulty estimation.  Furthermore, in Appendix D of the revised manuscript, we examine the correlation between problem-based difficulty and pass-rate difficulty, which exclusively reflects each model's problem-solving capabilities for each task. The moderate positive correlation suggests that problem-based difficulty effectively captures the inherent difficulty for the underlying LLMs of DS agents.
>
> 2) The Student agent retrieves examples from the built long-term memory based on task similarity, allowing for at least equivalent performance to traditional methods if the estimation is imprecise. Therefore, we can expect DSMentor to achieve performance on par with, if not superior to, traditional methods that do not consider the sequence of tasks.
>
> 3) Such costs of estimating difficulties are **offline, one-time investments**. The computed difficulties can be seamlessly applied to new incoming data or integrated with other datasets, potentially offsetting the need for additional expenditures.
> > [W3] *The order of retrieved examples in RAG has been already investigated in prior works (e.g., [1]); thus, it is not new for the community.*
>
> Thank you for pointing out this work. While both [1] and our work leverage RAG to enhance the performance of LLMs, the focuses are quite different. Specifically, our proposed method explores the construction of an (online) retrieval source (i.e., the pool of examples) automatically from previously tackled tasks, without relying on any external dataset. Furthermore, our approach emphasizes the importance of the order in which problems are tackled during inference, not just the order of retrieved examples. **We structure tasks in increasing order of difficulty and incorporate a growing long-term memory to retain prior experiences—an aspect that has not been investigated in prior works.**

---

> ### Author Response · Authors · 2024-12-01
> **Response to Reviewer XQT5 [Part 2]**
>
> > [W4] *The most exciting part of data science task is machine learning, which involves lots of details such as feature engineering, model selection, hyper-parameter tuning. However, this paper seems not to report any results on this.*
>
> Thank you for raising this concern. We evaluate DSMentor on DSEval [A1], which contains fundamental data processing concepts **as well as the machine learning tasks such as feature engineering, model fitting, and imbalanced dataset handling** (see the more knowledge points converge in Figure 9 of [A1]).
>
> [A1] Zhang, Yuge, Qiyang Jiang, Xingyu Han, Nan Chen, Yuqing Yang, and Kan Ren. "Benchmarking Data Science Agents." arXiv preprint arXiv:2402.17168 (2024)
>
> > [W5] *In Line 302-304, why is DSMentor run with three times? If temperature is set to 0 (which means greedy encoding), I cannot identify any randomness involved in this process.*
>
> Thank you for raising this question! In fact, a temperature of 0 indicates more determinism, but **zero temperature does not mean no randomness**. For instance, numerical deviation and breaking the tie could be nondeterministic. As [A2] investigated, LLMs with zero temperature also are more deterministic than warm temperatures, but still involve randomness. To this end, we conduct experiments three times and report the average results in our manuscript.
>
> [A2] Ouyang, Shuyin, et al. "LLM is Like a Box of Chocolates: the Non-determinism of ChatGPT in Code Generation." arXiv preprint arXiv:2308.02828 (2023).
>
> > [W6] *In Table 5 and Table 6, it seems that Random (Inc. Similarity) achieves competitive results compared with DSMentor, while avoiding extra inference computation for estimating difficulties.*
>
> Thank you for this question and careful review! It's true that in Tables 5 and 6, the Random (Increasing Similarity) approach yields competitive results compared to DSMentor, while our easy-to-hard curriculum design generally outperforms the Random approach across the four datasets in DSEval, the performance gap narrows for multiple-turn datasets, such as DSEval-Exercise and DSEval-Kaggle. However, each multiple-turn problem-set consists of several questions ranging from basic, easier tasks (e.g., reading a CSV file) to advanced, harder tasks (e.g., training a model). **This variety in question difficulty levels correspondingly diminishes the benefits of the curriculum-based order.**
>
> We acknowledge that the random approach avoids the additional computational overhead associated with estimating task difficulties. However, these costs are **offline, one-time investments**. The computed difficulties can be seamlessly applied to new incoming data or integrated with other datasets, potentially offsetting the need for additional expenditures.

---

### Official Review · Reviewer_y2UT · 2024-10-29

**Soundness:** 3
**Presentation:** 2
**Contribution:** 2
**Rating:** 5
**Confidence:** 4

**Summary:**

The paper introduces DSMentor, which uses the same LLM in Mentor and Student roles to curate and solve an inference curriculum for Data Science tasks. (problem, attempt, evaluation) triplets are accumulated online into a long-term memory, which can be retrieved from to bring the top-k most similar problems into context. These problems can then be ordered within the context in terms of increasing/decreasing similarity, or increasing/decreasing difficulty (according to the difficulty labels used to form the easy-to-hard problem curriculum).

**Strengths:**

The paper demonstrates that the method generally outperforms several appropriate baselines and shows that the design is well-motivated through comprehensive ablations. Overall, the experiments are thorough and their presentation is clear.

Using learning curricula in the context of data science problems is well-motivated, given the way that complex tasks can be decomposed into simpler ones.

The method is simple and appears to be effective.

**Weaknesses:**

AIDE [1] is a leading baseline for data science LLM agents that has not been considered.

In-context curricula for LLMs have been used in several other settings [2, 3], which brings into question the novelty of this work. Especially, given that nothing about the method seems data science specific, besides the difficulty scale guidelines.

[1] D. Schmidt, Z. Jiang and Y. Wu, AIDE: Human-Level Performance in Data Science Competitions, weco.ai, url: https://www.weco.ai/blog/technical-report,  2024

[2] Y. Liu, J. Liu, X. Shi, Q. Cheng, Y. Huang and W. Lu, Let’s Learn Step by Step: Enhancing In-Context Learning Ability with Curriculum Learning, arXiv preprint, arXiv:2402.10738, 2024

[3] H. Sun, L. Liu, J. Li, F. Wang, B. Dong, R. Lin and R. Huang, Conifer: Improving Complex Constrained Instruction-Following Ability of Large Language Models, arXiv preprint, arXiv:2404.02823, 2024

**Questions:**

The fact that the hard-to-easy curriculum does well on some tasks seems unintuitive; why do you think this is?

Why is the scope of the paper restricted to data science problems? Am I correct in my understanding that the method is largely domain-agnostic?

---

> ### Author Response · Authors · 2024-11-26
> **Response to Reviewer y2UT [Part 1]**
>
> Thanks for the reviewer’s valuable comments. We gratefully appreciate your acknowledgement of our evaluation and presentation, and we carefully reviewed all comments and addressed the raised concerns on a point-by-point basis, as follows.
>
>
> >[W1] *“AIDE [1] is a leading baseline for data science LLM agents that has not been considered.”*
>
> Thank you for bringing this work to our attention. In the revised version, we have included this paper as the related work to make our discussion on DS agents more comprehensive. However, we would like to address that while both [1] and DSMentor focus on the data science domain, these two works address different aspects. Specifically, [1] leverages performance feedback to iteratively refine code in a multi-turn fashion. In contrast, the proposed DSMentor focuses on **constructing an online, growing memory and retrieving examples from this memory** to help DS agents better understand and utilize fundamental skills. We believe that incorporating feedback to refine both the code and the difficulty design represents a promising direction for future work.
>
> >[W2] *In-context curricula for LLMs have been used in several other settings [2, 3], which brings into question the novelty of this work. Especially, given that nothing about the method seems data science specific, besides the difficulty scale guidelines.*
>
> Thank you for bringing these two papers to our attention. We acknowledge that both [2] and [3] highlight the importance of incorporating curriculum learning with LLMs, which aligns with our research interests. In the revised manuscript, we have included these two papers as related works to enrich the discussion on curriculum learning and LLMs. Although our work also utilizes curriculum learning for LLMs, we believe that our work offers **distinct contributions beyond the domain differences**. We would like to emphasize that **(1) the approaches of applying curriculum learning in [2] and [3] differ significantly from ours, and (2) these methods cannot be directly applied to data science agents.**
>
> More specifically, [2] focuses on ordering demonstrations retrieved from an offline, pre-fixed training set based on perplexity, whereas our approach emphasizes the ordering of inference tasks and the construction of a demonstration pool in an online manner. Notably, the perplexity-based approach in [2] requires the availability of all possible ground-truth reference codes during inference, making it unsuitable for code generation tasks.
>
> In addition, [3] develops an instruction-tuning dataset, Conifer, based on the ShareGPT dataset, which instructions of varying difficulty levels will be used for further training. This approach diverges from our focus on improving data efficiency during the post-training stage, where the curriculum-learning idea is applied to the ordering of data science tasks and information retrieval from an online, growing long-term memory.

---

> ### Author Response · Authors · 2024-11-26
> **Response to Reviewer y2UT [Part 2]**
>
> > [Q1] *The fact that the hard-to-easy curriculum does well on some tasks seems unintuitive; why do you think this is?*
>
> Thank you for the question and for the careful review! We believe the reviewer is referring to Table 5 of the ablation studies, where the hard-to-easy curriculum outperforms the easy-to-hard curriculum by 0.6% on the DSEval-Kaggle dataset. It is important to note that DSEval-Kaggle is a multiple-turn dataset in which each problem set consists of several questions ranging from basic, easier tasks (e.g., reading a CSV file) to advanced, harder tasks (e.g., training a model).
> In this context, the comprehensive nature of the problem sets, which encompass questions with diverse difficulty levels, diminishes the benefits or impacts of the curriculum task ordering. Instead, the similarities between problem sets may play a relatively more important role, which could explain why the hard-to-easy curriculum performs well on certain tasks.
>
> > [Q2] *Why is the scope of the paper restricted to data science problems? Am I correct in my understanding that the method is largely domain-agnostic?*
>
> Thank you for raising this question! Data science problems often involve vague task descriptions and complex data interpretation but can **be constructed by integrating simpler tasks** [A1], requiring a deep understanding of basic concepts and tasks to build effective solutions. Additionally, the availability of labeled or high-quality data in data science can be limited and the strict requirements of generating accurate results need up-to-data information, where the offline, external dataset may not be suitable or even not accessible. We address this motivation in the introduction (see Line 047-094 as well as Figure 1 in the manuscript), and we agreed that this approach has the potential to extend to other areas. For instance, it could be applied to domains such as software engineering, where tasks like debugging or code synthesis involve similar complexities and hierarchical problem-solving structures. Exploring these extensions would be a promising direction for future work.
>
>
> [A1] Sirui Hong, Yizhang Lin, Bangbang Liu, Binhao Wu, Danyang Li, Jiaqi Chen, Jiayi Zhang, Jinlin Wang, Lingyao Zhang, Mingchen Zhuge, et al. Data interpreter: An LLM agent for data science. arXiv preprint arXiv:2402.18679, 2024a

---

> > ### Comment · Reviewer_y2UT · 2024-11-30
> > **Thank-you for your response**
> >
> > Thank-you for including the new related works in your manuscript. Whilst I understand that there are distinctions between e.g., AIDE and DSMentor, it is still important to empirically compare to the performance of alternative frameworks that are also trying to improve the data science performance of LLMs in-context.
> >
> > Additionally, whilst the Conifer dataset is constructed for downstream fine-tuning, the role of the curriculum is in improving responses in-context, in much the same way as high quality responses obtained from your method could be aggregated into a synthetic training dataset.
> >
> > Finally, I appreciate that data science problems can be constructed by integrating simpler tasks, and that there may be limited suitable offline data, many reasoning and agentic tasks have these properties.

---

### Official Review · Reviewer_GGFc · 2024-10-30

**Soundness:** 2
**Presentation:** 3
**Contribution:** 1
**Rating:** 3
**Confidence:** 4

**Summary:**

This paper introduces DSMentor, a framework designed to improve the performance of large language model (LLM) agents in solving data science tasks through a combination of curriculum learning and long-term memory. The central innovation is organizing tasks in increasing order of difficulty (curriculum learning) and dynamically accumulating and retrieving knowledge from previous tasks (online memory). DSMentor is evaluated on two benchmarks: DSEval and QRData, where it outperforms baseline agents, particularly in causal reasoning tasks. The framework uses Claude-3.5-Sonnet and Llama-3.1-70b as base models and demonstrates performance gains  over existing methods.

**Strengths:**

1. DSMentor introduces an inference-time curriculum learning strategy that effectively mirrors human learning, gradually increasing task difficulty to improve the model’s problem-solving abilities.

2. The paper provides thorough experimentation across multiple benchmarks (DSEval, QRData), demonstrating consistent improvements in both general data science tasks and causal reasoning problems.

**Weaknesses:**

1. DSMentor’s performance may degrade when no similar questions exist in memory, and it lacks a clear strategy to handle novel tasks without prior examples. Additionally, as memory grows, retrieval efficiency may become an issue, but the paper doesn’t address how to manage this scalability challenge effectively.

2. The mentor agent’s ability to assign consistent difficulty ratings across diverse tasks is questionable. The generality of the Difficulty Scale Guidelines may lead to inaccurate assessments, especially for tasks that are too difficult for the mentor agent itself, risking hallucinated scores or poor curriculum design.

3. While the framework stores both correct and incorrect answers in memory, it’s unclear how incorrect answers contribute to future learning. This could lead to confusion if the model inadvertently learns from flawed data.

**Questions:**

1. How does DSMentor handle cases where no related or similar questions exist in the long-term memory? Would the model's performance degrade in such scenarios?

2. Why are traditional methods unable to effectively retrieve relevant datapoints from memory? What specific limitations do these methods have compared to DSMentor’s approach?

3. How large is the dataset used in DSMentor, and does each data point need manual labeling? If so, what is the process for labeling these data points, especially in large-scale datasets?

4. How does the mentor agent perform in generating difficulty scores across different tasks? Does it produce consistent and normalized difficulty ratings? Given the generality of the Difficulty Scale Guidelines in Figure 3, how does DSMentor ensure reliable and accurate difficulty assessments, especially for diverse task types?

5. The mentor agent’s effectiveness seems to rely on its ability to solve the task itself. What happens if the task is too difficult for the mentor agent, leading to incorrect or hallucinated answers and difficulty scores? How does DSMentor mitigate this risk?

6. As the long-term memory grows, how does DSMentor manage memory retrieval efficiency? What strategies are in place to prevent performance degradation when the memory size becomes too large?

7. What is the specific embedding model $ \varepsilon $ used for memory retrieval, and why is cosine similarity the metric of choice? How does this approach compare to other methods like Retrieval-Augmented Generation (RAG) for identifying relevant examples?

8. If the retrieved answer from long-term memory is incorrect, how does this contribute to improving future answers? Does DSMentor account for incorrect examples in a way that positively impacts performance?

---

> ### Author Response · Authors · 2024-12-01
> **Response to Reviewer GGFc [Part 1]**
>
> Thanks for the reviewer’s valuable comments and acknowledgement on the design of our framework and the “thorough experimentation”. We carefully reviewed your comments and please find the corresponding responses as follows.
>
> >[W1 & Q1] *“DSMentor’s performance may degrade ….” “How does DSMentor handle cases where no related or similar questions exist in the long-term memory? Would the model's performance degrade in such scenarios?”*
>
> Thank you for raising this concern. In the present work, we primarily focus on the data science domain, where the proposed framework retrieves tasks **from the same dataset that shares common features regarding data processing**. According to Tables 3 and 4 in the manuscript, DSMentor, augmented with retrieved examples, demonstrates **increased the pass rate by 3.3%-9.8% compared to baselines that lack such examples**. Additionally, in the current design of DSMentor, the Student agent is informed that the examples are sorted by increasing similarity and is prompted to **summarize useful information from the provided examples before generating code**, to mitigate the potential impacts of unnecessary information. We also want to address that the utility of data retrieval based on the similarity of text embedding has been explored in prior research like [A1] to enhance performance of agents, while we believe that integrating with a similarity threshold or classifying the tasks before inference could further enhance the robustness of the proposed approach in the scenarios without related examples.
>
> >[W1 & Q6] *“Additionally, as memory grows, retrieval efficiency may become an issue, but the paper doesn’t address how to manage this scalability challenge effectively.” “As the long-term memory grows, how does DSMentor manage memory retrieval efficiency? What strategies are in place to prevent performance degradation when the memory size becomes too large?”*
>
> We acknowledge that as memory grows, it becomes more important to achieve retrieval efficiency, while the main purpose of this paper is to enhance the problem-solving capabilities of data science agents. There are multiple potential solutions used in real-world applications that can be incorporated to improve the retrieval efficiency summarized in [A2]. For instance, we could periodically examine the utility of stored examples and prune older or less frequently accessed examples to keep the memory size manageable like the truncated memory, which is also used in Reflexion [A3].
>
> >[W2 & Q4] *“The mentor agent’s ability to assign consistent difficulty ratings across diverse tasks is questionable.”; “How does the mentor agent perform in generating difficulty scores across different tasks? Does it produce consistent and normalized difficulty ratings? Given the generality of the Difficulty Scale Guidelines in Figure 3, how does DSMentor ensure reliable and accurate difficulty assessments, especially for diverse task types?”*
>
> Thank you for raising this concern. We would like to clarify that this work focuses on data science agents and correspondingly, the difficulty scale guidelines are specifically tailored for the data science domain. Our evaluation contains different types of DS tasks as described in Line 286-293 of the manuscript, where DSMentor shows outstanding performance across different tasks.
>
> In addition, we compared the effectiveness of problem-based difficulty, which utilizes the difficulty scale guidelines, with other difficulty metrics in Section 4.4.1. From Table 3 and 4, we observe that using the problem-based difficulty **outperforms that with the manual difficulty** (based on the official difficulty and pass rate provided by LeetCode) by 5% for DSMentor-Llama-3.1-70b and achieves the same pass rate for DSMentor-Claude-3.5-Sonnet, which substantiates the effectiveness of difficulty estimation. Furthermore, in Appendix D of the revised manuscript, we examine the correlation between problem-based difficulty and pass-rate difficulty, which exclusively reflects each model's problem-solving capabilities for each task. The moderate positive correlation suggests that problem-based difficulty effectively captures the inherent difficulty for the underlying LLMs of DS agents.

---

> ### Author Response · Authors · 2024-12-01
> **Response to Reviewer GGFc [Part 2]**
>
> >[W2 & Q5] *“The generality of the Difficulty Scale Guidelines may lead to inaccurate assessments, especially for tasks that are too difficult for the mentor agent itself, risking hallucinated scores or poor curriculum design.”; “The mentor agent’s effectiveness seems to rely on its ability to solve the task itself. What happens if the task is too difficult for the mentor agent, leading to incorrect or hallucin bibated answers and difficulty scores? How does DSMentor mitigate this risk?”*
>
> We would like to clarify that in our work, the Difficulty Scale Guidelines serve as system prompts that assist in the difficulty estimation process and is tailored for data science tasks, while the Mentor agent will **further refine this estimation by considering the perspective of involved concepts, algorithmic complexity, and implementation challenges** to reason out the difficulty score. (See appendix B.1 for more details.)
>
> We acknowledge that both the Mentor and Student agents utilize the same base model. However, compared to fully solving the task, the process of identifying the required skills based on the features of the problem description is generally considered easier. This process guides the Student agents to structure the long-term memory in a sequence of progressively increasing difficulty. Such a simpler process, coupled with additional guidance, expands the range of problems that the Student agent can effectively tackle, thereby enhancing the robustness against difficult tasks.
>
>
> >[W3 & Q8] *“While the framework stores both correct and incorrect answers in memory, ….”;“If the retrieved answer from long-term memory is incorrect, how does this contribute to improving future answers? Does DSMentor account for incorrect examples in a way that positively impacts performance?”*
>
> Thank you for raising this concern. In Section 4.4.3, we examined the impact of adding incorrect examples to the memory and whether they contribute to future learning. As shown in Table 7, we observed that enabling access to incorrect examples **does positively impact the performance**, particularly for datasets with fewer problem sets (e.g., DSEval-LeetCode and DSEval-Exercise). However, for datasets with a larger number of problem sets (e.g., DSEval-SO, DSEval-Kaggle, and QRData), the improvement from adding incorrect examples is marginal. We conjecture that incorrect examples may offer useful context and signals through problem prompts and environment feedback, helping to offset the negative effects of limited data availability/diversity and thereby improve future answers.
> This observation aligns with findings in prior work, such as [A4], which demonstrates the effectiveness of learning from both positive and negative feedback. By incorporating both correct and incorrect answers, the framework facilitates an iterative refinement process that helps improve performance on more challenging tasks.

---

> ### Author Response · Authors · 2024-12-01
> **Response to Reviewer GGFc [Part 3]**
>
> [Q1] See the response to [W1&Q1] in Part 1.
>
> >[Q2] *“Why are traditional methods unable to effectively retrieve relevant datapoints from memory? What specific limitations do these methods have compared to DSMentor’s approach?”*
>
> Thank you for the question. The key limitation of traditional methods lies in their **inability to utilize the inherent difficulty relationships among tasks to guide agents in gradually enhancing their problem-solving abilities**.  In Section 4.5.2, we compare the proposed method with traditional approaches that retrieve and rank examples solely based on similarity. The primary distinction lies in the construction of the memory pool: our method includes only previously solved examples that are easier than the current task. This design enables agents to more effectively understand and utilize simpler examples to tackle relatively harder questions.
> As shown in Tables 5 and 6, both DSMentor-Llama-3.1-70b and DSMentor-Claude-3.5-Sonnet outperform random curricula (i.e., the traditional methods that do not consider the sequence of the tasks) when using either the easy-to-hard curriculum. These results highlight the efficacy of our curriculum-based approach in enhancing the agent’s performance.
>
> >[Q3] *How large is the dataset used in DSMentor, and does each data point need manual labeling? If so, what is the process for labeling these data points, especially in large-scale datasets?*
>
> We would like to clarify that in Line 286-293 of the manuscript, we mentioned that DSMentor is evaluated on diverse datasets, including DSEval-LeetCode (with 40 single-turn problem-sets), DSEval-StackOverflow (with 202 single-turn problem-sets), DSEval-Exercise (with 21 multiple-turn problem-sets), and DSEval-Kaggle (with 31 multiple-turn problem-sets), and QRData (with 411 statistical and causal reasoning problems).
>
> Manual labeling is not required for the dataset used in DSMentor, as the difficulty of each task is assessed automatically following the difficulty guidelines based on the problem prompts.
>
> [Q4]  See the response to [W2&Q4] in Part 1.
>
> [Q5]  See the response to [W2&Q5] in Part 1.
>
> [Q6] See the response to [W1&Q6] in Part 1.
>
> >[Q7] *What is the specific embedding model ε used for memory retrieval, and why is cosine similarity the metric of choice? How does this approach compare to other methods like Retrieval-Augmented Generation (RAG) for identifying relevant examples?*
>
> Thanks for your question. We would like to clarify that in Line 299-300 of the manuscript, we mention that this work uses the cohere.embed-english-v3 as the embedding model to retrieve relevant examples from memory, **optimized specifically for cosine similarity**. Cosine similarity is also a widely used metric in Retrieval-Augmented Generation (RAG) systems, as used in [A1, A5]. Our method retrieves examples based on their similarities, ranking them by increasing similarity.
>
> A key distinction of our approach compared to traditional methods is **the composition of our example pool**. Specifically, while RAG typically includes a wide range of examples from easier to more challenging tasks from offline corpus as [A1], our method constructs a long-term memory that contains only previously solved examples that are not more difficult than the current task without additional datasets. Such retrieval could provide additional guidance for solving current tasks based on previously built skills.
>
> [Q8]  See the response to [W3&Q8] in Part 1.
>
> **Reference**
>
> [A1] Zhang, Lei, Yuge Zhang, Kan Ren, Dongsheng Li, and Yuqing Yang. "MLCopilot: Unleashing the Power of Large Language Models in Solving Machine Learning Tasks." In Proceedings of the 18th Conference of the European Chapter of the Association for Computational Linguistics (Volume 1: Long Papers), pp. 2931-2959. 2024.
>
> [A2] Zhao, Siyun, Yuqing Yang, Zilong Wang, Zhiyuan He, Luna K. Qiu, and Lili Qiu. "Retrieval Augmented Generation (RAG) and Beyond: A Comprehensive Survey on How to Make your LLMs use External Data More Wisely." arXiv preprint arXiv:2409.14924 (2024).
>
> [A3] Shinn, Noah, Federico Cassano, Ashwin Gopinath, Karthik Narasimhan, and Shunyu Yao. "Reflexion: Language agents with verbal reinforcement learning." Advances in Neural Information Processing Systems 36 (2024).
>
> [A4] Liu, Hao, Carmelo Sferrazza, and Pieter Abbeel. "Chain of Hindsight aligns Language Models with Feedback." In The Twelfth International Conference on Learning Representations (2024).
>
> [A5] Liu, Jiachang, Dinghan Shen, Yizhe Zhang, Bill Dolan, Lawrence Carin, and Weizhu Chen. "What Makes Good In-Context Examples for GPT-3?." DeeLIO 2022 (2022): 100.

---

### Official Review · Reviewer_SRBx · 2024-11-04

**Soundness:** 3
**Presentation:** 3
**Contribution:** 3
**Rating:** 5
**Confidence:** 5

**Summary:**

This paper presents an inference-time optimization framework called DSMentor for large language model (LLM) agents, focusing on enhancing performance by addressing the overlooked aspect of problem-solving order during inference. By leveraging curriculum learning, DSMentor introduces tasks in ascending order of complexity. Evaluation of DSEval and QRData benchmarks reveals the performance improvements in causal reasoning tasks compared to GPT-4. This work emphasizes the importance of effective knowledge accumulation and utilization during inference, mirroring human learning and paving new ways to optimize LLM performance through curriculum-based approaches.

**Strengths:**

(1) The paper demonstrates the outstanding performance of DSMentor, which utilizes an easy-to-hard curriculum, across established benchmarks in data analysis and causal reasoning tasks.

(2) The paper shows that by retrieving easier and more relevant examples from memory and structuring the knowledge in an increasing-similarity order, DSMentor significantly improves data science agents' understanding.

(3) The paper demonstrates how DSMentor incrementally introduces tasks with increasing complexity, facilitating the agent's progressive learning of causal relationships.

**Weaknesses:**

(1) While the paper presents promising results, it would be more compelling if the experiments were repeated multiple times to establish a mean performance. This would provide a clearer picture of the consistency and reliability of the proposed method, DSMentor, across different runs and potentially highlight any variability in its performance.

(2) A more comprehensive exploration of the shortcomings, including the reasons behind them and possible mitigation strategies, would strengthen the paper's contributions and provide valuable insights for future research.

**Questions:**

None.

**Details Of Ethics Concerns:**

None.

---

> ### Author Response · Authors · 2024-11-26
> **Response to Reviewer SRBx**
>
> Thanks for the reviewer’s valuable comments and the acknowledgement on the outstanding performance of our proposed framework. We carefully reviewed your comments and addressed them on a point-by-point basis, as follows.
>
> >[W1] *“While the paper presents promising results, it would be more compelling if the experiments were repeated multiple times to establish a mean performance. This would provide a clearer picture of the consistency and reliability of the proposed method, DSMentor, across different runs and potentially highlight any variability in its performance.”*
>
> Thank you for highlighting this aspect. We agree that consistency and reliability are important factors to evaluate the performance of DSMentor. In fact, this suggestion aligns with our experimental setups. Specifically, we **conduct the experiments at least three times with a temperature set to 0** to minimize randomness as much as possible, and we report the average results in our paper.
>
> >[W2] *“A more comprehensive exploration of the shortcomings, including the reasons behind them and possible mitigation strategies, would strengthen the paper's contributions and provide valuable insights for future research.”*
>
> Thank you for your suggestion. In the proposed framework, the long-term memory is constructed in an online fashion and grows by increasing difficulty, enabling the Student agent to better understand and utilize examples/demonstrations more effectively. However, there are several promising directions to enhance DSMentor’s generalizability and flexibility. We mentioned them in Line 536-540 of our manuscript and would like to elaborate them as follows. Currently, examples are retrieved from the long-term memory based on the similarity of problem prompts, which may overlook underlying knowledge connections. To address this, we could **leverage prior/expertise-based knowledge to retrieve examples** within the same or related categories, ensuring greater relevance between examples and the current task. Additionally, instead of predefining the curriculum-based dataset before solving tasks, we could **incorporate detailed feedbacks from the environment** (e.g., error messages) to dynamically refine difficulty designs and correct generated code when necessary. We believe that these improvements could further strengthen the framework’s adaptability and effectiveness.

---

### Meta-Review · Area_Chair_JZef · 2024-12-23

**Metareview:**

This manuscript introduces an inference-time optimization framework called DSMentor for large language model (LLM) agents, with a goal of enhancing agent performance through a combination of curriculum learning and long-term memory. The key innovation is dynamically accumulating and retrieving knowledge from previous tasks (online memory) and organizing retrieved tasks in-context in an increasing order of difficulty (curriculum learning). Evaluation of DSMentor is done on two benchmarks: DSEval and QRData, where it outperforms baseline agents, particularly in causal reasoning tasks. The framework uses Claude-3.5-Sonnet and Llama-3.1-70b as base models and demonstrates performance gains over existing methods.

While all reviewers agree that the idea is intuitive, easy to understand, and the experiments are thorough and presentation clear, there are concerns raised regarding the reliability of such a method. Specifically, it's unclear whether performances will deteriorate when randomness (seed, sampled temperature) is introduced. DSMentor’s performance may also degrade when there's a lack of similar questions stored in memory, aka when the model encounters a truly novel, "test" problem. Moreover, the performance hinges on the  mentor agent’s ability to assign consistent difficulty ratings across diverse tasks, which is also not fully guarantee-able.

The paper could benefit from more thorough analyses on its failure scenarios, its reliability, and scalability. We look forward to seeing the improvement versions of this work in future submissions.

**Additional Comments On Reviewer Discussion:**

During the rebuttal period the authors diligently responded to all reviewers, however only one reviewer ended up further engaging with the authors. The others fell silent, despite multiple reminders from the AC. No reviewers had their scores changed.

---

### Decision · Program_Chairs · 2025-01-22

Reject